# Dust Formation in the Wind of AGB Stars—The Effects of Mass, Metallicity and Gas-Dust Drift

**Silvia Tosi** [1,2], **Flavia Dell'Agli** [2] and **Erendira Huerta-Martinez** [2] and **Paolo Ventura** [2,3,*]

1   Dipartimento di Matematica e Fisica, Universitá degli Studi Roma Tre, Via della Vasca Navale 84, 00100 Rome, Italy; silvia.tosi@uniroma3.it
2   Istituto Nazionale di Astrofisica, Osservatorio Astronomico di Roma, Via Frascati 33, 00078 Monte Porzio Catone, Italy; flavia.dellagli@inaf.it (F.D.); eren.huerta@gmail.com (E.H.-M.)
3   Istituto Nazionale di Fisica Nucleare, Section of Perugia, Via A. Pascoli snc, 06123 Perugia, Italy
*   Correspondence: paolo.ventura@inaf.it

**Abstract:** Dust production in the wind of stars evolving through the asymptotic giant branch is investigated by using a stationary wind model, applied to results from stellar evolution modelling. Results regarding 1–8 $M_\odot$ stars of metallicities $Z = 0.014$ (solar) and $Z = 2 \times 10^{-3}$ are compared, to infer the role played by stellar mass and chemical composition on the dust formation process. We find a dichotomy in mass: stars of (initial) mass below $\sim$3 $M_\odot$ produce silicates and alumina dust before they become carbon stars, then carbonaceous dust; the higher mass counterparts produce only silicates and alumina dust, in quantities that scale with metallicity. The presence of drifts with average drift velocities $\sim$5 Km/s leads to higher dust formation rates owing to the higher growth rates of the dust grains of the different species. However, no significant changes are found in the overall optical depths, because the higher rate of dust formations favours a fast expansion of the wind, that prevents further significant production of dust. As far as oxygen-rich stars are concerned, the presence of drifts makes the main dust component to change from olivine to pyroxene. The release of the assumption that the number density of the seed particles is independent of the dust species considered affects dust formation in the wind of carbon stars: a factor 10 reduction in the density of the seeds of SiC leads to bigger sized SiC grains, and partly inhibits the formation of solid carbon, since the wind is accelerated and the densities in the carbon formation zone are smaller. No substantial differences are found in the winds of oxygen-rich stars.

**Keywords:** stars: AGB and post-AGB; stars: evolution; stars: abundances





## 1. Introduction

Stars of initial mass in the 1–8 $M_\odot$ range after the exhaustion of central helium evolve through the asymptotic giant branch (AGB) phase, which preceeds the general contraction that leads to the white dwarf evolution [1]. AGB stars have been recognized as efficient dust manufacturers, owing to the thermodynamic conditions of their cool end dense winds, that prove an extremely favourable environment for the deposition of gaseous molecules into solid particles [2]. Understanding dust production by AGB stars is therefore crucial for a number of hot topics in modern Astrophysics, connected to the overall dust budget in the Universe. The presence of dust grains in the surroundings of stars can deeply modify the spectral energy distribution, that is shifted into the mid-IR [3] and is characterised by distinguished spectral features, related to the specific dust species formed. The interpretation of IR observations of resolved stellar populations requires the knowledge of the amount and the mineralogy of the dust present in the circumstellar envelope of the individual sources.

To face these challenges some research groups have recently coupled stellar evolution with dust formation, by implementing the description of the formation and growth of dust grains in the winds of evolved stars, according to the variation of the fundamental

parameters of the star during the evolution. The first works where results from stellar evolution were used to model dust formation across the AGB lifetime and calculate the dust yields produced by stars of different mass and metallicity were published by [4–7]. These findings have been extensively used to determine the global dust production of the stars during their AGB life and the overall dust production rate of galaxies from AGB stars: the early studies on this regard were focused on the Magellanic Clouds [8], followed by investigations of dust production in Local Group galaxies [9,10]. Further steps forward came when the studies on dust production were coupled with radiative transfer modelling, opening the possibility to predict the evolution of the spectral energy distribution (SED) of the stars while evolving through the AGB and to produce synthetic photometry. This was the starting point to characterize the evolved stellar populations of the Magellanic Clouds [11–14] and of some galaxies belonging to the Local Group [10,15], in terms of mass and formation epoch of the progenitors. The most recent and detailed works on this field were published by [16,17], who studied in detail hundreds of sources in the Large Magellanic Cloud, and confronted the synthetic SED obtained by stellar evolution + dust formation modelling with the observed spectra taken with the IRS instrument onboard the Spitzer Space Telescope: the analysis by [16,17] allowed an exhaustive characterization of the sources investigated, with the determination of several trends connecting the mineralogy of the dust in the surroundings of the star and the dust formation rate, with the mass and metallicity of the progenitors.

The methodology used to model dust formation relies on the schematization proposed by the Heidelberg group, where a stationary wind is allowed to expand from the surface of the star, until reaching the dust formation zone, where it is accelerated by the effects of radiation pressure on the newly formed dust grains [2,18–20]. Dust particles and gas molecules are assumed to travel with the same speed until reaching an asymptotic velocity, with which the material is dispersed into the interstellar medium. The solid particles are assumed to grow onto pre-existing seeds, whose number density is proportional to the density of hydrogen molecules: the scaling factor has been so far chosen as independent of the dust species considered.

In this contribution we present the most recent models for dust formation in the winds of AGB stars and discuss the efficiency of dust production by stars of solar metallicity. To outline the role of metallicity we compare these findings with metal-poor models of metallicity $Z = 2 \times 10^{-3}$. We discuss some limitations of the current description of dust formation, and focus on the the role of the gas-dust drifts and on possible effects of differences among the number densities of the seeds of different dust species.

The paper is structured as follows: the methodology and the numerical codes used to model AGB evolution and dust formation in the wind of AGBs are described in Section 2; a general overview of the evolution of stars across the AGB is given in Section 3; dust production by AGB stars is discussed in Section 4; the role of gas-dust drift is described in Section 5, whereas the effects of the choice of the seed densities of the various dust species are discussed in Section 6; finally, the conclusions are given in Section 7.

## 2. Stellar Evolution and Dust Formation Modelling

The evolutionary sequences presented in this contribution were obtained by means of the ATON code for stellar evolution, which is described in detail e.g., in [21], where the interested reader can find an accurate analysis of the numerical structure and of the input physics adopted. Here we recall the physical ingredients most relevant for the description the advanced evolutionary phases of stars, and of the AGB phase itself.

Nuclear burning and mixing of chemicals are self-consistently coupled by means of a diffusive scheme, following the prescription by [22]. The temperature gradient with regions unstable to convective motions is found via the full spectrum of turbulence model [23]. The rate of mass loss during the phases when the star is oxygen-rich at the surface is modelled via the treatment of [24]; for the carbon star phases, we used the results on mass loss published from the Berlin group [25,26]. The surface molecular opacities were calculated

via the AESOPUS tool [27], which allows us to consider the effects of the variation of the surface abundances of the CNO species.

Convective eddies are allowed to enter regions of the star stable against convective motions, by imposing an exponential decay of velocities beyond the formal border, found via the Schwarzschild criterion. We did not consider further mixing mechanisms, such as those proposed in the recent works by [28,29], who outlined the importance of Magnetohydrodynamics-based mixing, particularly during the inwards penetration of the convective envelope after each thermal pulse (TP). The inclusion of these mixing mechanisms in the description of stellar evolution has a relevant impact on the yields of the heavy nuclei and to understand the details of the isotope distribution of different species, which can be tested, e.g., via the analysis of presolar grains. However, the overall gas yields of the different chemical species and the predictions regarding the dust budget that AGB stars produce, on which this paper is focused, can be safely determined by adopting a basic description of overshoot, such as the one used in the present work.

We calculated stellar models of initial mass in the 1–8 $M_\odot$ range, with metallicities $Z = 0.014$ and $Z = 2 \times 10^{-3}$, and helium mass fractions $Y = 0.268$ and $Y = 0.25$, respectively. The former corresponds to the solar metallicity [30], whereas the latter, corresponding approximately to [Fe/H] $\simeq -1$, is the typical chemical composition of metal-poor, old disc stellar populations. All the evolutionary sequences were started from the pre-MS phase and proceed until the almost complete ejection of the convective envelope, after the beginning of the general contraction, which starts the post-AGB evolution.

Dust formation was modelled according to the method proposed by [2]. The stationary outflow is assumed to expand radially from the photosphere of the star, until entering the region of dust formation, where it is accelerated by the effects of radiation pressure on dust grains. Use of the stationary approximation for the outflow stems from the fact that when comparing the velocity and density profiles of such stationary winds with published models of dust-forming pulsators, one observes a strong resemblance in most cases to the average velocity and density profiles on which the outwards propagating shocks are superposed. We can therefore assume that the estimate of the quantities of dust formed in the outflow based on such an average outflow structure is reasonable [2].

The dynamics of the wind is described via the momentum conservation equation:

$$v\frac{dv}{dr} = -\frac{GM}{r^2}(1-\Gamma)$$
(1)

where $v$ and $M$ indicate the gas velocity and the mass of the star at a given evolutionary stage, respectively. $\Gamma$ indicates the relative effect of the radiation pressure against the gravitational pull, and is given by the expression

$$\Gamma = \frac{kL}{4\pi cGM}$$
(2)

$L$ is the stellar luminosity, while $k$, in cm$^2$/gr unit, indicates the overall extinction coefficient, which is approximated by the expression

$$k = k_{\text{gas}} + \sum_i f_i k_i$$
(3)

The sum in Equation (3) is extended to all the dust species considered, each characterized by an extinction coefficient $k_i$. $k_{\text{gas}}$ represents the contribution to the extinction from gas molecules, that becomes negligible out of the region from which the formation of dust begins. The $f_i$'s in Equation (3) represent the fraction of the chemical species most relevant for the formation of a given dust compound (the so-called "key species", after [2]), that is condensed into dust:

$$f_i = \frac{4\pi(a_i^3 - a_{i,0}^3)}{3V_{i,0}}\frac{n_{d,i}}{\epsilon_i n_H}$$
(4)

$a_{i,0}$ and $a_i$ in Equation (4) indicate the initial and the current size of the grains of the *i*-th dust species, $V_{i,0}$ is the volume of the monomer, while $\epsilon_i$ is the abundance of the key species at the surface of the star, relative to hydrogen. Dust grains are assumed to grow on pre-existing seeds, of number density $n_{d,i}$, in turn proportional to the hydrogen density, $n_H$. The study of how the choice of the proportionality constant between seeds and hydrogen density affects the results related to dust production is one of the objectives of the present work. The grain size distribution is found to peak towards the smallest scales in the regions close to the deposition zone, whereas the external regions of the circumstellar envelope are populated by bigger size particles; eventually an asymptotic size is reached, as the gas densities drop to values too small to favour further growth of the grains.

It goes without saying that for an exhaustive description and comprehension of the formation of dust in the wind of evolved stars, a more complete treatment, based on nucleation theory, is required, to determine the nature and the density of the micro-particles that act as seeds for the growth of dust grains. Use of an apriori seed density might affect the description of the wind and of dust production in the cases when little quantities of dust are produced; conversely, in the situations where dust is formed efficiently, the action of the radiation pressure makes the wind to experience a strong acceleration, which favours the rapid achievement of the asymptotic conditions mentioned above. Because the majority of the dust produced by AGB stars is formed during the afore mentioned phases characterized by large dust production rates, the determination of the dust yields can be safely calculated by relying on the approximations adopted in the present work. The assumption of a single size distribution of the dust grains at a given radial distance from the surface of the star might affect the predictions regarding the expected SED, but are not relevant for the estimates of the dust production rates and of the dust yields.

The dust species considered are silicates and alumina dust ($Al_2O_3$), in the winds of oxygen-rich stars, while solid carbon and silicon carbide (SiC) are taken into account for what attains carbon stars. Solid iron is formed in either cases. The silicate species treated explicitly are olivine ($Mg_{2x_{ol}}Fe_{2(1-x_{ol})}SiO_4$), pyroxene ($Mg_{x_{ol}}Fe_{1-x_{ol}}SiO_3$) and quartz ($SiO_2$). In olivine and pyroxene the magnesium and iron components coexist in a single species; $x_{ol}$ and $x_{py}$ give the fraction of the Mg components in olivine and pyroxene, respectively. The Mg components are generally more stable [2], thus they are found to overcome the Fe components in either species. Both $x_{ol}$ and $x_{py}$ decrease as the formation of silicates takes place (see Equations (12) and (16) in [2]), but they exceed 70% in all cases investigated so far. The key species for the different dust particles are: (a) silicon, for silicates and silicon carbide; (b) aluminium, for $Al_2O_3$; (c) carbon, for solid carbon; (d) iron, for solid iron. In agreement with [2] we assume that for all the dust species considered $a_{i,0} = 10^{-3}$ μm and $n_{d,i} = 10^{-13}n_H$.

On the mathematical side the model is composed by the differential equations describing the behaviour of velocity (Equation (1) above) and of the optical depth, complemented by the equations giving the growth rate of the grains of the various dust species and the radial profile of the gas density, that is found via the mass conservation law:

$$\rho(r) = \frac{\dot{M}}{4\pi r^2 \mathrm{v}} \tag{5}$$

where $\dot{M}$ is the mass-loss rate experienced by the star. All the relevant equations are extensively described in [2]. The description of the dust formation process requires the following ingredients, which are found via stellar evolution modelling: effective temperature, mass, luminosity and the surface chemical composition, this last required for the determination of the $\epsilon_i$'s.

The knowledge of the radial distribution of the density of the gas and of the size of dust particles are used to calculate the optical depth of the circumstellar envelope, which in the present study will be considered at the wavelength $\lambda = 10$ μm:

$$\tau_{10} = \int \pi n_d Q_{10} a^3 dr \tag{6}$$

The integral on the right hand side of Equation (6) is extended over the region of the circumstellar envelope where dust particles form and grow. $Q_{10}$ is the extinction coefficient associated to a given dust species at 10 µm, while $n_d$ and $a$ are the density of the seed particles over which the grains form and the grain size, respectively. Equation (6) must be applied to all the dust species formed.

## 3. AGB Evolution: A General Overview

The evolution of AGB stars is primarily affected by the initial mass of the star, which proves the key parameter in determining the time scale of the AGB evolution and the modification of the surface chemistry. The latter can be altered by two physical mechanisms, which produce two different chemical patterns. The third dredge-up (TDU) is experienced when each thermal pulse is extinguished, and consists in the inwards penetration of the internal regions of the surface convection to stellar zones previously processed by helium burning [31]: the main effect of TDU is the gradual enrichment in the surface carbon, which might eventually lead to the formation of carbon stars. Hot bottom burning (HBB) consists in the activation of proton capture nucleosynthesis at the bottom of the convective envelope during the inter-pulse phases [32]. The ignition of HBB requires core masses of the order of $\sim$0.8 M$_\odot$ [33], which reflects into initial masses of the star of the order of $\sim$3 M$_\odot$. The main effects of HBB are the synthesis of large quantities of nitrogen at the expenses of carbon. In metal-poor environments the depletion of the surface oxygen and magnesium and the synthesis of aluminium take place [34]. The occurrence of HBB prevents the formation of carbon stars, as the surface carbon nuclei are exposed to proton captures. According to this discussion, the evolution of AGB stars can be broadly divided into two main categories, according to whether the initial mass is below or above the threshold limit required to activate HBB. This is the path that we will follow in the discussion of the properties of stars of different mass.

### 3.1. The Evolution towards and after the Carbon Star Stage

An example of the AGB evolution of a low-mass star not experiencing HBB is shown in Figure 1, which reports the time variation of the main physical and chemical properties of a 1.5 M$_\odot$ star of solar metallicity. The results in the left panel of the figure show the gradual increase in the luminosity of the star, due the increase in the core mass as the hydrogen burning shell moves outwards (in mass) [35]. The effects of the various TDU events can be seen in the sudden rise in the surface C/O ratio, shown in the right panel of Figure 1. The surface carbon enrichment and the achievement of the carbon star stage favour a significant increase in the surface molecular opacities [36], which causes a large expansion of the star, a decrease in the surface gravity and the increase in the mass loss rate [37,38]. This can be seen in the run of the total mass of the star in the right panel of Figure 1 (red line in the figure), where it is evident that the decrease in the total mass becomes much faster after the star becomes a carbon star, which in this specific case occurs after the last TP experienced. The large sensitivity of the surface molecular opacities to the carbon content of the envelope somewhat prevents the accumulation of surface carbon in excess of $\sim$1% of the overall mass fraction, as the loss of the envelope inhibits the occurrence of further TDU episodes.

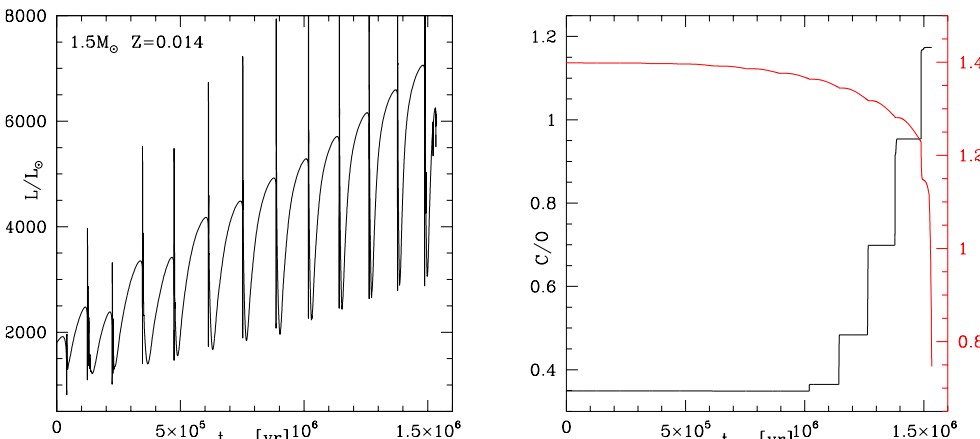

**Figure 1.** (**Left**) Time variation of the luminosity of a 1.5 M$_\odot$ model star of metallicity Z = 0.014, during the AGB phase. (**Right**) Time variation of the surface C/O ratio (black line, scale on the left) and of the mass of the stars (red track, scale on the right) of the same model star shown in the left panel.

### 3.2. Evolution and Nucleosynthesis of Stars Undergoing Hot Bottom Burning

The evolution of the stars experiencing HBB is different from their lower mass counterparts, both on the physical and the chemical side. An example of this kind of evolution is reported in Figure 2, which refers to a 4 M$_\odot$ model star of solar metallicity. The luminosity of the star (see left panel) initially follows an increasing trend, which is partly associated to the growth of the core mass and also to the ignition of HBB, which is known to lead to a significant increase in the overall luminosity, such that the stars exposed to HBB deviate from the classic core mass vs luminosity relationship [32].

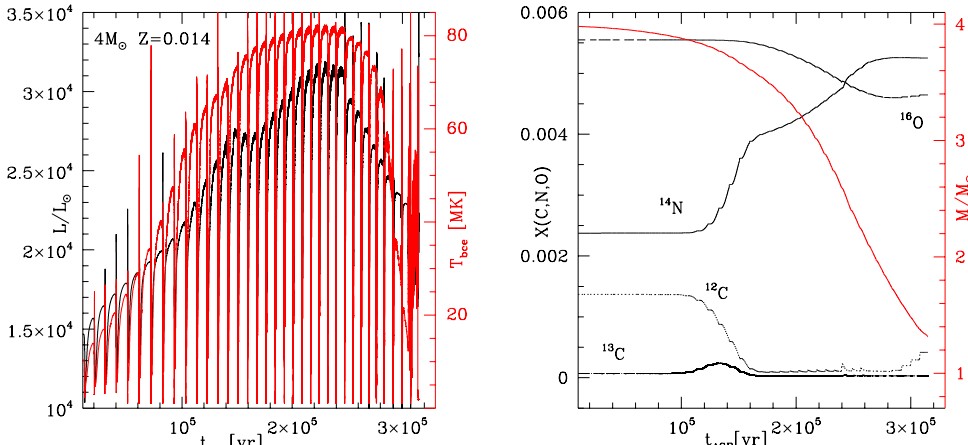

**Figure 2.** (**Left**) Time variation of the luminosity (black, scale in the left) and of the temperature at the bottom of the convective envelope (red, scale on the right) of a 4 M$_\odot$ model star of metallicity Z = 0.014, during the AGB phase. (**Right**) Time variation of the surface $^{12}$C (dotted line), $^{13}$C (dotted-dashed line), $^{14}$N (solid), $^{16}$O (dashed), and of the mass of the stars (red track, scale on the right) of the same model star shown in the left panel.

During the second part of the evolution the luminosity decreases, because the loss of the external mantle causes the gradual extinction of HBB. The behaviour of the luminosity is tightly connected with the trend followed by the temperature at the base of the envelope (T$_{bce}$, red line in the left panel of Figure 2), which in this specific case reaches a maximum of ~80 MK, then decreases during the final evolutionary phases. For these stars the rate of

mass loss is strongly dependent on the luminosity of the star [24]: the mass loss process is more gradual than in their lower mass counterparts investigated previously, reaching a maximum rate in conjunction with the largest luminosity. This is confirmed by the mass vs time relationship shown in the right panel of Figure 2 (red line).

The variation of the surface chemistry, shown in the right panel of Figure 2, reflects the signature of HBB, which in this case is seen to start after $\sim 10^5$ yr since the beginning of the AGB phase. The ignition of HBB leads to the depletion of the surface $^{12}$C (dotted track) in favour of $^{13}$C (dotted-dashed track) and nitrogen (solid line). Soon after the start of HBB the surface $^{12}$C/$^{13}$C tends to the equilibrium value, which is between 3 and 4, the precise value being sensitive to the temperature at the bottom of the envelope. Some oxygen depletion (dashed line) also takes place, starting during the phase when $T_{bce}$ is highest.

### 3.3. The Role of Mass and Metallicity on the AGB Evolution

To highlight the role of mass and metallicity on the AGB evolution, we show in Figure 3 the variation of the luminosity of model stars of different mass, and metallicity Z = 0.014 (left panel) and $Z = 2 \times 10^{-3}$ (right). The time reported in the abscissa is in a logarithmic scale, to allow to show all the evolutionary sequences on the same plane.

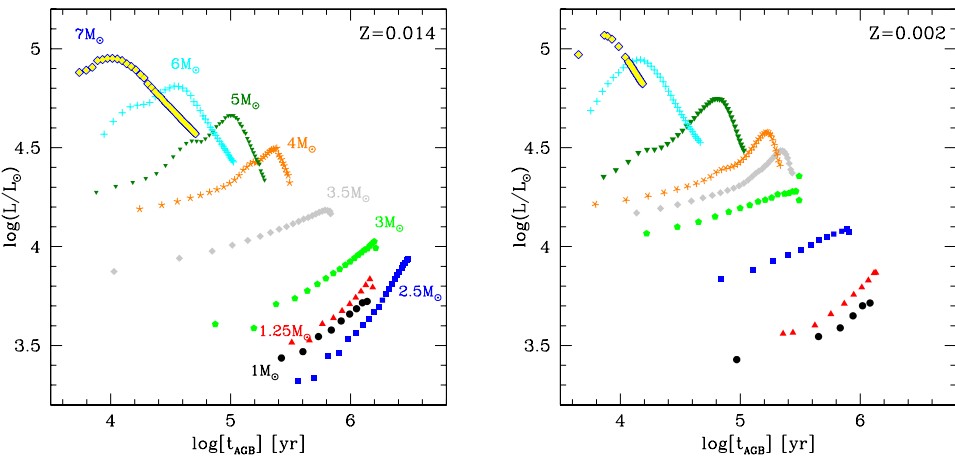

**Figure 3.** (**Left**) Time variation of the luminosity of solar metallicity model stars during the AGB phase. Each point corresponds to an inter-pulse phase. The numbers near the different tracks and symbols indicate the initial mass of the star. Times are counted from the beginning of the AGB phase. (**Right**) Same as the left panel, but for the $Z = 2 \times 10^{-3}$ metallicity. The same correspondence between the symbols and the initial mass of the star was adopted.

For both metallicities it is evident the increase in the luminosity at which the stars evolve through the AGB with the initial mass: this is connected to the larger core mass attained by the stars of higher mass after the core helium burning phase, which is particularly evident in the M > 2 M$_\odot$ mass domain, where helium burning takes place under quiescent conditions; conversely, the lower mass counterparts develop electron degeneracy during the ascending of the red giant branch, thus they begin the AGB phase with core masses of similar mass. We note the different behaviour of the luminosity between low mass stars, which become brighter and brighter as they evolve on the AGB, and the stars experiencing HBB, whose luminosity, after reaching a maximum value, decreases during the final evolutionary phases, for the reasons discussed above.

The difference in the luminosity among stars of different mass reflects into the time scale of the AGB evolution, which becomes shorter and shorter as the mass increases: massive AGBs experiencing strong HBB evolve in $\sim 50$ kyr, whereas the duration of the AGB phase of low mass stars is in some cases above 1 Myr. In the low mass domain the trend of the duration of the AGB phase with mass is not monotonic. On the one hand the lower the mass of the star, the lower the luminosity, the longer the evolutionary time-scale; however, this effect is counterbalanced by the little mass initially present in the envelope,

which makes the evolutionary time scales shorter. This is the reason why the stars for which the AGB phase reaches the longest duration are those with initial mass $\sim$ 2–2.5 M$_\odot$.

The comparison between the results reported in the two panels of Figure 3 shows that lower metallicity stars reach higher luminosities for a given initial mass, as they evolve on more massive cores [33]. Metallicity also affects the minimum threshold mass required to ignite HBB, which decreases from 3.5 M$_\odot$, in the solar metallicity case, to 3 M$_\odot$, for $Z = 2 \times 10^{-3}$.

The difference between the behaviour of the stars experiencing HBB and that of the lower mass counterparts is particularly evident in the variation of the surface chemical composition. This is shown in Figure 4, which reports the variation of the surface carbon (top panels) and oxygen (bottom panels) for the same model stars shown in Figure 3. The surface carbon of M $\leq$ 3 M$_\odot$ stars increases during the AGB, owing to the action of repeated TDU episodes, until reaching abundances of the order of $\sim$0.01; the higher the initial mass, the larger is the final mass fraction of carbon at the surface of the star, because the stars of higher mass experience a higher number of TPs (hence of TDU events) before the envelope is lost. The surface carbon content reached during the final evolutionary phases is approximately independent of the metallicity, as the carbon dredged-up to the surface after each TP is of primary origin, being produced in the helium burning shell.

In the stars experiencing HBB the surface carbon is destroyed by proton capture reactions taking place at the base of the convective envelope. This process starts when the temperature in those regions of the stars exceeds $\sim$30 MK [39], a condition which is met earlier during the AGB the larger the mass of the star. The carbon depletion is of the order of 1 dex, fairly independently of mass and metallicity: this is because full activation of HBB leads to a chemical composition characterised by equilibrium CNO abundances, which for the case of carbon corresponds to a factor $\sim$10 depletion factor. During the final AGB phases the surface carbon increases, after HBB was turned off by the consumption of the external envelope, leaving room to TDU to rise the surface carbon content.

The different behaviour of low mass and massive AGBs can be seen also in the evolution of the surface $^{16}$O, shown in the bottom panels of Figure 4. The increase in the surface $^{16}$O favoured by the action of TDU in low mass stars is smaller than for $^{12}$C, since most of the material dredged-up to the surface is made up of $^{12}$C. The variation of the surface oxygen of the stars experiencing HBB is extremely sensitive to the metallicity: while in the solar metallicity model stars (left panel) the $^{16}$O is seen to diminish by not more than a factor $\sim$2, in the metal-poor case the oxygen reduction factor reaches one order of magnitude: this is due to the large sensitivity of the strength of HBB to the metallicity of the stars, discussed in detail by [34].

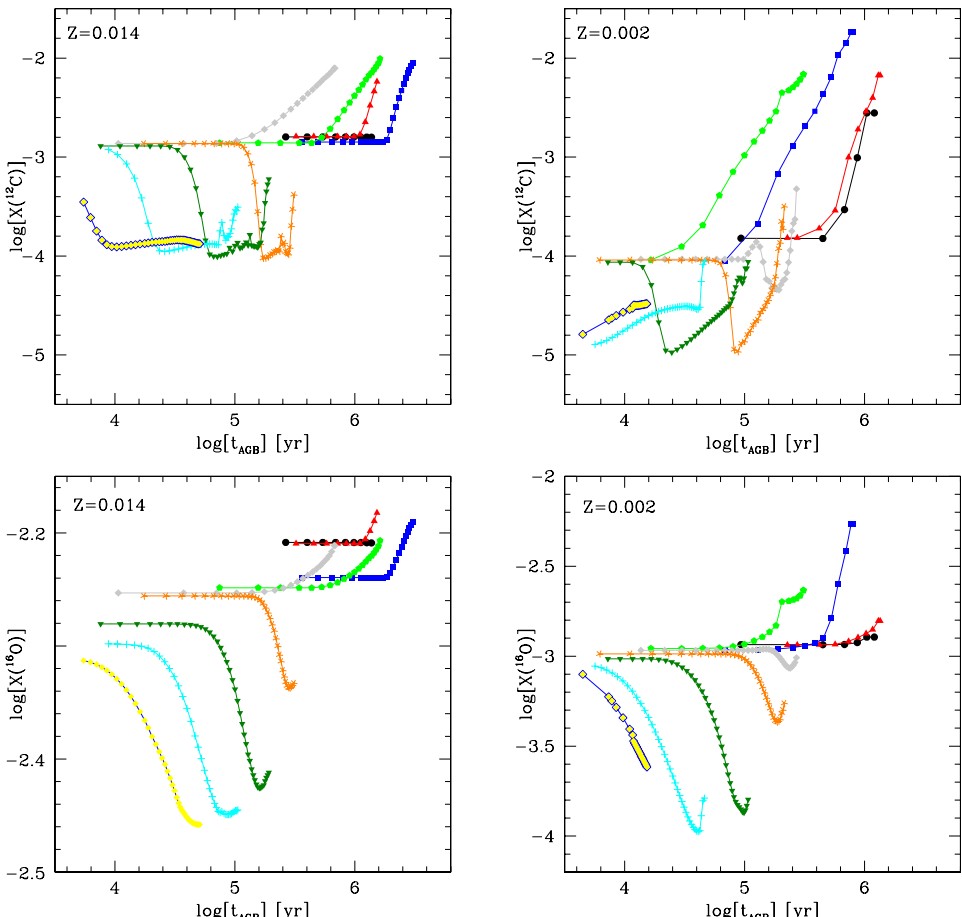

**Figure 4.** Time variation of the surface carbon (**top** panels) and oxygen (**bottom** panels) mass fractions of the same model stars reported in Figure 3. The same meaning of the different symbols was adopted. Both times and surface abundances are reported in logarythmic scales.

## 4. Dust Production by AGB Stars

The evolution of the physical and chemical properties of the stars is tightly related to the dust formation process in the wind of the stars during the various AGB evolutionary phases. The mineralogy of the dust produced is mainly determined by the surface C/O, with carbon stars producing solid carbon and SiC particles, whereas oxygen-rich stars produce silicates and alumina dust.

On the bases of the discussion of the previous section we deduce that dust formation in $M > 3 \, M_\odot$ stars is limited to the formation of silicates and alumina dust, whereas low mass stars produce silicates and $Al_2O_3$ during the first part of the evolution, then carbonaceous species after the achievement of the carbon star stage.

The formation region of the dust particles is determined by the degree of stability of the various species, which in turn is related to the formation enthalpies of the solid compounds [2]. In carbon rich environments the most stable species is silicon carbide, which forms $\sim$2–3 stellar radii ($R_*$) from the surface of the star, at temperatures of the order of 1400 K; the growth of solid carbon grains starts when the temperature drops below 1100 K, typically $\sim$7–10 $R_*$ from the photosphere [40]. The winds of oxygen-rich are characterized by a similar situation, with alumina dust particles formed in a more internal region of the circumstellar envelope, whereas silicates form in a more external region, at temperatures not exceeding 1050 K [40]. Silicon carbide and alumina dust, despite the higher stability, form in smaller quantities compared to solid carbon and silicates; this is due to the lower availability of the relevant key species, namely silicon (for SiC) and

aluminium (for $Al_2O_3$), compared to the corresponding key species of solid carbon (carbon) and silicates (silicon). For this reason, and for the generally lower extinction coefficients of SiC and $Al_2O_3$ with respect to solid carbon [2] and silicates, the acceleration of the gas in the wind is mostly favoured by the formation of the latter species. The large efficiency of the formation of SiC and $Al_2O_3$ make these species to reach saturation conditions. In the SiC case this is obtained once the fraction of gaseous silicon reaches 55% (note that 45% of the gaseous silicon is absorbed into the highly stable SiS molecules [2]); as for alumina dust, saturation is reached when all the gaseous aluminium is locked into dust particles.

Figure 5 shows the variation of the DPR during the evolution of the same model stars reported in Figure 3. The results shown in Figure 5 can be understood by considering the tight link between the DPR and the mass loss rate, as the latter determines the density stratification of the wind (see Equation (5)), thus the number of gaseous molecules available to form dust. Massive AGBs of solar metallicity reach peak DPRs that span the $10^{-7}$–$10^{-6}$ $M_\odot$/yr range: the larger the mass of the star the higher the mass loss rate experienced, which favours larger DPRs [5]. Most of the DPR is due to the formation of silicates, with alumina dust contributing for less than 10%; this is due to the lower availability of gaseous aluminium in the surface regions of the stars with respect to silicon. The DPR reached by metal-poor massive AGBs are between 5 and 10 times smaller than their solar metallicity counterparts, due to the lower silicon mass fractions, related to the lower metallicity.

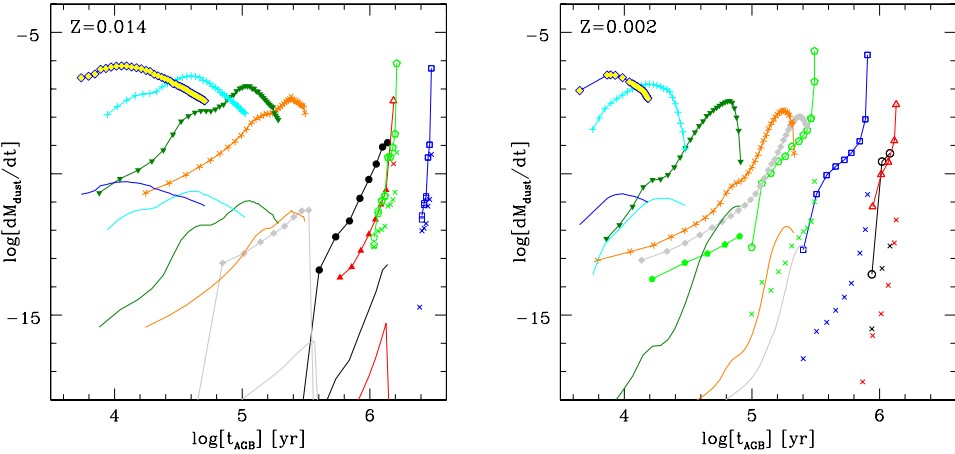

**Figure 5.** (**Left**) Time variation of the dust production rate of model stars of different initial mass and solar metallicity. The same symbols as in Figures 3 and 4 were adopted to indicate the stars of different initial mass. Full points refer to dust composed by silicates and alunina dust, while open points indicate carbonaceous dust. Thin solid lines and crosses indicate the contribution from alumina dust and SiC dust, respectively. (**Right**) Same at the left panel, for the $Z = 2 \times 10^{-3}$ metallicity.

In low mass stars not experiencing HBB the dust production rate during the first part of the AGB lifetime, mostly due to the formation of silicate particles, is significantly smaller when compared with the corresponding DPR of the higher mass counterparts, because the mass loss rates are between one and three orders of magnitude lower. On the other hand these stars experience very large DPRs during the final AGB phases, during which they are carbon rich, as the accumulation of large quantities of carbon favours a considerable expansion of the stars, which lowers the surface gravity and enhances the mass-loss rates, hence the DPR. The values of the DPRs reached at the end of the AGB evolution exceed $10^{-6}$ $M_\odot$/yr, corresponding to the peak DPR experienced by massive AGBs. The progeny of 2–3 $M_\odot$ stars were identified as the counterparts of the stars exibiting the large infrared excess in the Large Magellanic Cloud [13,14]. The contribution of SiC is between 10% and 20% in the solar metallicity case, whereas it can be considered negligible in the $Z = 2 \times 10^{-3}$ stars, because the overall silicon locked in the star approximately scales with metallicity, thus it is significantly smaller than in higher $Z$ stars.

An exception to the general rule that dust production by low-mass AGBs during the phases when they are oxygen rich is negligible, is represented by the stars that fail to achieve the C-star stage. An example of this class of objects is represented by the 1 $M_\odot$ model star of solar metallicity, whose behaviour is shown in the left panel of Figure 5. The DPR experienced by this star at the end of the AGB phase is higher than the DPRs of the stars experiencing soft HBB (as the 3.5 $M_\odot$ star, indicated with grey diamonds in the same panel of the figure) and is also much larger than the DPR reached by low-mass stars of the same metallicity during the oxygen-rich phase. This behaviour is explained by the increase in the mass loss rate that takes place during the final AGB phases, which causes an increase in the DPR. These stars should be at the origin of presolar $Al_2O_3$ grains recovered in meteorites [41]. A thorough understanding of the origin of these particles requires considering the effects of non-standard stellar mixing in the calculation of the evolutionary sequences [28]; however, as stated in Section 2, this is far beyond the scopes of the present investigation.

The overall dust yields by the model stars considered in the present investigation are shown in Figure 6. The quantity indicated with black diamonds is the total mass of dust produced during the whole AGB phase of a star of a given initial mass (reported on the abscissa). For the reasons given above this is dominated by silicates and solid carbon; the contribution by alumina and SiC dust are also indicated, with green triangles and red squares, respectively.

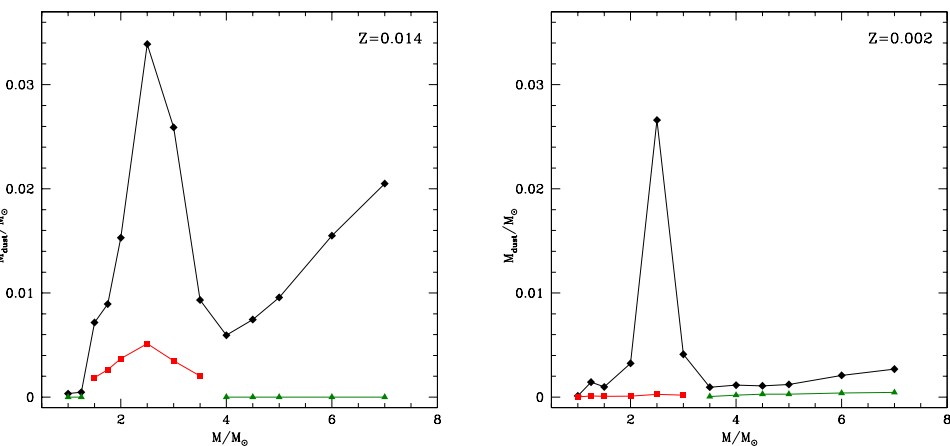

**Figure 6.** The total mass of stars with the initial mass reported on the abscissa is indicated with black diamonds for the metallicities $Z = 0.014$ (**Left**) and $Z = 2 \times 10^{-3}$ (**Right**). Red squares and green triangles indicate the contribution due to SiC and alumina dust, respectively.

In the $M \leq 2.5\,M_\odot$ mass domain the dust mass produced increases with the initial mass of the star; this dust is mainly composed by solid carbon, and is sensitive to the amount of carbon accumulated in the surface regions of the star by TDU, which we have shown to be higher the higher the mass of the star. The largest masses of dust, of the order of $0.03\,M_\odot$, are produced by $2.5\,M_\odot$ stars.

The carbon dust is higher in the solar metallicity case than in the metal-poor case, because higher metallicity stars assume a more expanded configuration, thus experience larger mass loss rates, which favour dust formation. The contribution from SiC dust is around 10% in the solar metallicity case, whereas it is negligible at $Z = 2 \times 10^{-3}$. This results is consistent with the recent study by [42], who showed that AGB stars descending from 2–2.5 $M_\odot$ progenitors, with metallicity close to solar, are the main parents of SiC presolar grains. In particular, SiC grains from these stars can account for the so called Main-Stream component [43], that based on their isotopic ratios of various chemical species are established to be produced by AGB stars [44].

For what attains massive AGBs, producing mostly silicates, the mass of dust produced during the AGB lifetime increases with the mass of the star, because the higher the initial

mass of the star the larger the mass loss rates experienced, thus the dust formation rate and the amount of dust formed. Note that in the solar metallicity case the dust mass produced by the 7 $M_\odot$ model star, of the order of 0.02 $M_\odot$, is smaller than the amount of carbon dust produced by the 2.5 $M_\odot$ model star. This result, coupled with the use of any realistic mass function, indicates that the overall dust from stellar populations is dominated by carbon dust, even in solar metallicity environments.

The dust mass produced by metal-poor massive AGB is significantly smaller that in the solar metallicity case, once more owing to the scarcity of silicon and aluminium in the external regions of the star.

## 5. Dust-Gas Drift

The results presented so far were obtained on the basis of the assumption of full coupling between dust and gas in the wind, which move with the same velocity. In this section we explore the effects of a drift velocity $w$ of dust grains with respect to the gas. We study the effects of a drift following the approach by [45]. The growth rate $J_{gr}$ of dust grains is multiplied by a factor that is sensitive to the drift velocity. The final expression is

$$J_{gr} = V_0 \alpha n_{gr} \sqrt{\frac{kT}{2\pi m_{gr}}} \sqrt{1 + \frac{\pi \mu m_H}{8kT} w^2} \tag{7}$$

where $m_{gr}$ and $n_{gr}$ are the mass and number density of the dust particles, $V_0$ is the volume of the monomer on which deposition begins, $\alpha$ is the sticking coefficient, $\mu$ and $T$ are the molecular weight and temperature of the gas where dust grains form and grow, $w$ is the drift velocity. The additional factor on the right hand side of Equation (7) becomes unity when no drift is considered ($w = 0$).

Figure 7 compares the results presented in the previous sections with those obtained by considering the effects of drifts. The latter are found to be negligible when drift velocities of the order of 1 Km/s are considered, thus we restrict the comparison to the no drift vs. $w = 5$Km/s cases. The top panels of Figure 7 refer to low-mass stars that reach the C-star stage, represented by the 1.5 $M_\odot$ and 3 $M_\odot$ cases. The bottom panels focus on massive AGBs that experience HBB, with the 4 $M_\odot$ model representing the stars experiencing soft HBB, and the 6 $M_\odot$ model representing the brighter counterparts, which are exposed to strong HBB conditions at the base of the envelope (see Figure 3).

In the top panels we chose the current mass of the star as indicator of the evolutionary stage, to be able to describe in more details the final phases during which the stars rapidly reach an extremely large infrared excess, before starting the post-AGB contraction. Conversely, in the bottom panels we use the AGB time on the abscissa.

On general grounds, the presence of drift rises dust production, as the growth rate of the dust particles, given by Equation (7), increases. In the 1.5 $M_\odot$ case we find that the formation rate of silicates during the first part of the AGB phase, before the C-star stage is reached, is higher in the drift case, with the fraction of silicon condensed into dust rising from ∼2% to ∼15%. During the following phase, after the star becomes a C-star, the drift slightly affects the production of solid carbon, with the fraction of carbon condensed into solid particles rising from ∼8% to ∼11%. No effects on the formation of SiC dust are found, as saturation conditions are reached since the early C-star phases, with the fraction of silicon condensed into SiC reaching the largest possible value of 55%, as discussed in Section 4.

For what concerns the 3 $M_\odot$ model star, we limit the discussion to the C-star phase, because as discussed in Section 4 little dust production takes place during the oxygen-rich phase. The production of carbon dust is generally higher when drift is considered, the largest discrepancy in the fraction of gaseous carbon condensed into dust being of the order of 30%. The results reported in Figure 7 show that SiC production is higher when drift is taken into account during the initial AGB phases, as saturation conditions are reached more easily in presence of drift (see Equation (7)). On the other hand no difference is found

after the mass of the star drops below ∼2.2 M$_\odot$, after saturation of silicon occurred in the standard case with no drift.

We see in the top-right panel of Figure 7 that despite the higher dust production found in the models with drift, the optical depth $\tau_{10}$ (see definition via Equation (6)) is not significantly affected. This is because the enhancement of the growth rate of the dust grains determines a faster acceleration of the wind, which based on Equation (5) causes a fast drop of the gas densities, thus preventing significant further formation of dust. The increase in the optical depth connected to the effects of drift during the evolution of carbon stars are within ∼10%.

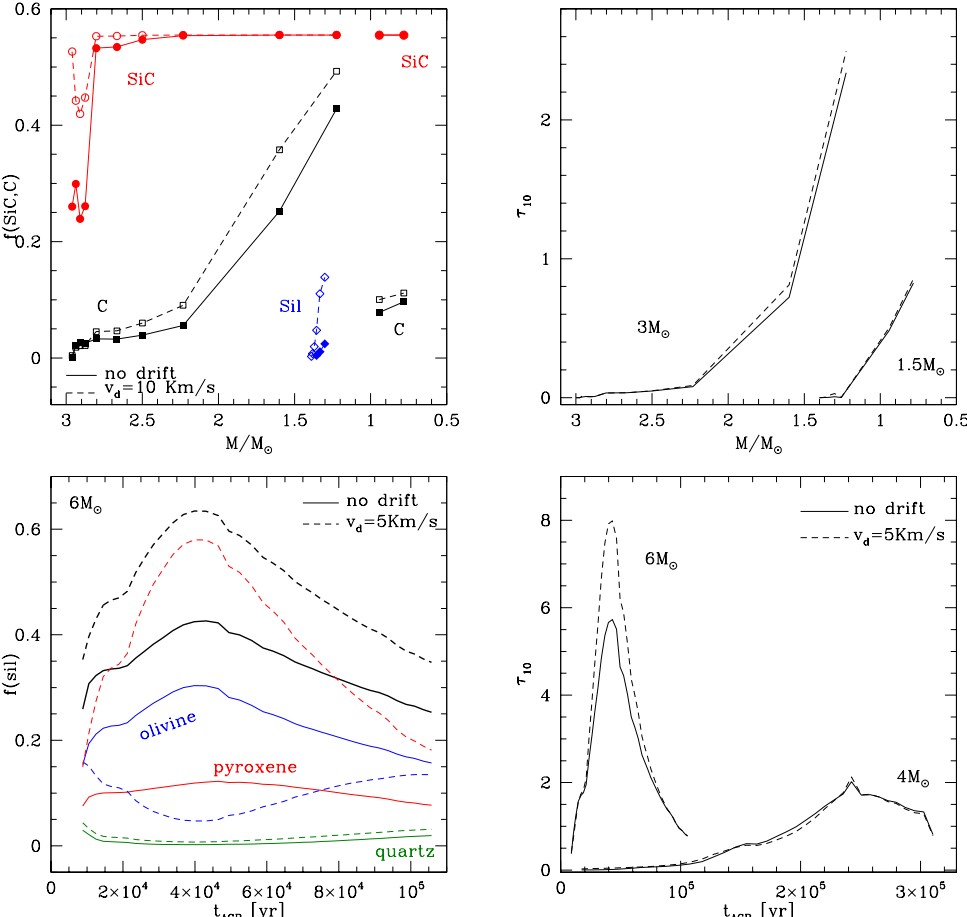

**Figure 7. Top, left**: Fraction of gaseous carbon condensed into dust (black), of silicon condensed into SiC (red) and silicates (blue) for the model stars of initial mass 1.5 M$_\odot$ and 3 M$_\odot$. The results are shown as a function of the current mass of the star. Full points connected with solid lines and open points connected with dashed tracks refer to results obtained with no drift and with drift velocities 5 Km/s, respectively. **Top, right**: Variation of the optical depth during the AGB phase of the same model stars shown in the top, left panel. **Bottom, left:** Time variation of the overall fraction of silicon condensed into dust (black lines), and of the individual fractions condensed into olivine (blue), pyroxene (red) and quartz (green), during the AGB phase of a 6 M$_\odot$ model star. Solid and dashed tracks refer to results obtained without drifts and with drift velocities 5 Km/s, respectively. **Bottom, right**: Time variation of the optical depth for the model stars of initial mass 4 M$_\odot$ and 6 M$_\odot$, obtained without drift (solid lines) and with drift velocities 5 Km/s (dashed).

Turning to massive AGBs experiencing HBB, we see in the left bottom panel of Figure 7 that the action of drift, similarly to low-mass stars, affects the fraction of silicon condensed into dust, which is on the average ∼30% higher in the model taking drift into account; the largest discrepancy, approaching ∼50%, is found in conjuntion with the phases of largest luminosity and dust production. However, the most striking result evident in Figure 7

is the effects of drift on the dust mineralogy. In the standard case with no drift most of the silicon is condensed to form olivine, whereas the contribution of pyroxene and quartz account for $\sim$25% and $\sim$3%, respectively. When drift is considered, most of the silicon is condensed to form pyroxene, with olivine and quartz contributing for $\sim$15% and $\sim$2%, respectively.

The reason for this difference is in the hybrid nature of olivine and pyroxene, discussed in Section 2, with both species being composed by a generally dominating magnesium component, coexisting with an iron component [2]. In the standard case with no drift the magnesium components account for $\sim$90% of olivine and pyroxene (these are the values attained by the $x_{ol}$ and $x_{py}$ quantities discussed in Section 2), a condition which favours the formation of olivine dust with respect to pyroxene. When drift is considered, the growth rate of the dust grains increases, which according to the discussion in Section 2 favours the decrease of both $x_{ol}$ and $x_{py}$, with the equilibrium of both the species shifting towards higher percentages of the iron component, which now accounts for $\sim$30% of the total of olivine and pyroxene. Under these conditions, pyroxene turns out to be most stable species, thus providing the dominant contribution to the formation of silicates. For readability we show in the bottom, left panel of Figure 7 only the 6 $M_{\odot}$ case; however the conclusions given above hold generally for all the stars exposed to HBB.

As the inclusion of drift favours dust production, we expect that the optical depth reached by the stars are found to be higher when drift is accounted for. In the stars experiencing soft HBB, represented by the 4 $M_{\odot}$ model star in the bottom, right panel of Figure 7, little difference is found between the results obtained with and without drift. On the other hand in the 6 $M_{\odot}$ case, where the optical depths attained are generally higher, considering drift leads to a $\sim$30% increase in the largest optical depth reached.

## 6. The Role of Differential Seed Density

The common assumption of dust modelling using the schematization of the Heidelberg group [2] is that the grains of all the dust species taken into account grow on nanometer sized seeds, characterized by the same number density, taken as $10^{-13} n_H$, where $n_H$ is the number density of hydrogen atoms. The only exception to this general assumption is found in the works by [6,7], who assumed that the number density of the seeds upon which solid carbon grains grow is dependent on the carbon excess with respect to oxygen. To understand how differences in the number density of the seeds of the different dust species might affect the results obtained, particularly those discussed in the previous sections, we investigated the effects of assuming that the number density of the seeds of SiC dust particles, $n_d(\text{SiC})$, are a factor of 2 and 10 smaller than those of solid carbon; the same was done for the study of dust production in the winds of oxygen-rich stars, by assuming the same scaling factor for the seed density of $Al_2O_3$ grains, $n_d(Al_2O_3)$.

The results obtained by changing the seed density of SiC particles in the circumstellar envelope of carbon stars are shown in Figure 8, where in the different panels we report the AGB evolution of the largest size reached by solid carbon and SiC grains (left panel), the fraction of silicon and carbon condensed into SiC and solid carbon, respectively, and the overall optical depth at 10 $\mu$m. The model stars on which this exploration is based are the same shown in the top panels of Figure 7. Even in this case we use the current mass of the star as indicator of the evolutionary status, as the extremely fast increase in the dust production rate that takes place during the very final AGB phases does not recommend to use time.

As a general rule, the choice of assuming a smaller number density of seed particles leads to lower quantities of dust of the corresponding species. This general behaviour holds as far as saturation conditions are not reached; there is no room to alter the dust quantity formed otherwise. In the case of SiC, we showed in the previous sections that saturation occurs at a given stage during the C-star phase, as all the silicon survived to the formation of SiS molecules, i.e., $\sim$55%, is condensed into SiC dust. The results shown in the middle panel of Figure 8 show that this saturation conditions is reached in all the model stars discussed;

the only effect of the choice of $n_d(SiC)$ on the results is that the SiC production rate is smaller in the models based on smaller $n_d(SiC)$'s, thus the saturation conditions of SiC are reached in a later phase with respect to the standard case with $n_d(SiC)/n_d(H) = 10^{-13}$.

The choice of the seed density affects the size reached by the dust grains, because the same fraction of silicon condensed into SiC is obtained by larger size SiC grains the smaller is the assumed seed number density. This is confirmed by the results shown in the left panel of Figure 8, where we see that the size of SiC grains reaches 0.1 µm, 0.13 µm and 0.22 µm in the $n_d(SiC)/n_d(H) = 10^{-13}$, $5 \times 10^{-14}$ and $10^{-14}$ cases, respectively.

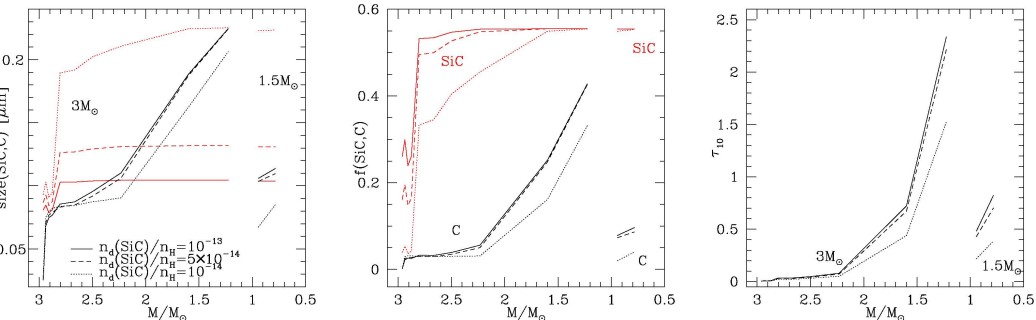

**Figure 8.** (**Left**) Variation of the size of silicon carbide (red) and solid carbon (black) grains formed in the wind of stars if initial mass 1.5 M$_\odot$ and 3 M$_\odot$ during the AGB phase. The current mass of the star is reported on the abscissa. The solid lines correspond to the assumption that the seed density of SiC grains, $n_d(SiC)$, is equal to that of solid carbon particles (solid line), whereas the dotted and dashed lines refer to results obtained by assuming $n_d(SiC) = 10^{-14}n_H$ and $n_d(SiC) = 5 \times 10^{-14}n_H$, respectively. (**Middle**) Variation of the fraction of silicon condensed into SiC (red lines) and of the fraction of gaseous carbon condensed into solid carbon grains (black) for the same models shown in the left panel. (**Right**) Variation of the optical depth $\tau_{10}$ of the 1.5 M$_\odot$ and 3 M$_\odot$ model stars reported in the left and middle panels.

The size of the carbon grains formed and the fractions of gaseous carbon condensed into dust are substantially independent of the choice of $n_d(SiC)$ in the $5 \times 10^{-14} \leq n_d(SiC)/n_d(H) \leq 10^{-13}$ range; conversely, when $n_d(SiC)/n_d(H) = 10^{-14}$ the size of carbon grains drops to 0.22 µm (to be compared to 0.25 µm found for the other cases) and the fraction of carbon condensed into dust is between 10 and 30% smaller. These results can be explained by focusing on the dynamics of the winds of carbon stars, as they move out from the photosphere, driven by radiation pressure. In the standard case examined so far we find that the acceleration of the wind of carbon stars is determined by the formation of solid carbon. In the more internal regions where SiC grains form and grow the radiation pressure is generally too low to favour the acceleration of the wind: this is related to the lower quantities of SiC formed, in comparison with solid carbon, and to the extinction coefficients of SiC, which are significantly smaller than those of solid carbon [2]. The results regarding the solid carbon dust in the $n_d(SiC)/n_d(H) = 10^{-13}$ and $5 \times 10^{-14}$ cases are extremely similar, because the formation of SiC has no significant effects in the dynamics of the wind and does not affect the formation of solid carbon grains. On the other hand, in the $n_d(SiC)/n_d(H) = 10^{-14}$ case, SiC grains grow sufficiently big to cause the acceleration of the wind before the formation of carbon dust takes place: therefore the densities in the region where formation of carbon grains takes place are smaller, so are the size reached by carbon grains and the fraction of gaseous carbon absorbed into dust. These results also affect the optical depth, shown in the right panel of Figure 8, which is smaller in the lowest $n_d(SiC)$ case; the differences concerns both the 1.5 M$_\odot$ and 3 M$_\odot$.

We did not explore the effects of the choice of higher densities of the seeds of SiC dust. On the other hand, based on the results, we may safely assume that the results would not change as far as the formation of carbon dust and thus of the overall optical depth are concerned: indeed in this case we expect the formation of smaller size SiC grains, which

will not be able to favour any acceleration of the wind, thus leaving the conditions of the formation zone of carbon grains unchanged.

The effect of the choice of assuming different seed densities for alumina dust and silicates in the wind of oxygen-rich stars was explored by considering the cases $n_d(Al_2O_3)/n_H = 10^{-14}$ and $5 \times 10^{-14}$; these results are compared with the standard case $n_d(Al_2O_3)/n_H = 10^{-13}$. We apply these changes to the 4 $M_\odot$ and 6 $M_\odot$ model stars, shown in the bottom panels of Figure 7. The results of this exploration is shown in Figure 9, where we report the time variation of the size reached by olivine and $Al_2O_3$ grains and of the fraction of silicon and aluminium condensed into silicates and alumina dust, respectively.

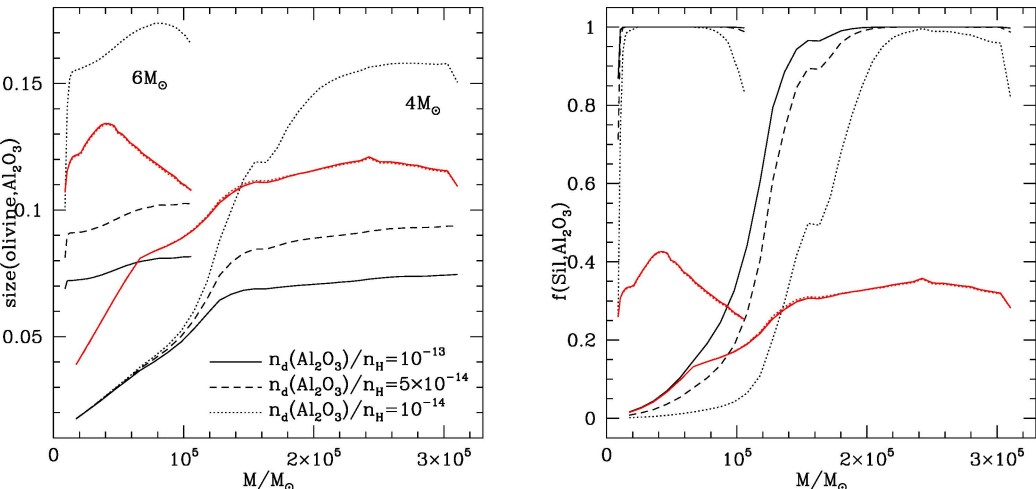

**Figure 9.** The AGB variation of the size of olivine (red lines) and $Al_2O_3$ particles (**Left**) and of the fraction of silicon and aluminium condensed into silicates and alumina dust, respectively (**Right**), of the model stars of initial mass 4 $M_\odot$ and 6 $M_\odot$. The different lines refer to the choices regarding the seed density of $Al_2O_3$ grains.

Similarly to the investigation presented earlier in this section on the role played by the choice of the seed density of SiC dust, we find that the smaller $n_d(Al_2O_3)$ the larger the size reached by alumina dust particles. In the left panel of Figure 9 we see that the largest sizes attained by alumina dust particles increase from 0.07 μm, in the standard case examined previously, to 0.1 μm for $n_d(Al_2O_3)/n_H = 5 \times 10^{-14}$, to 0.17 μm for $n_d(Al_2O_3)/n_H = 10^{-14}$. This is once more due to saturation effects, as the large stability of $Al_2O_3$ favours the formation of this dust species close to the photosphere of the star, thus easing saturation of the gaseous aluminium in the wind. Still consistently with previous discussions we find that the onset of saturation of aluminium is postponed to later phases when smaller $n_d(Al_2O_3)$'s are assumed. This is seen to happen particularly in the 4 $M_\odot$ model star. In the 6 $M_\odot$ case thus formation is so efficient since the early AGB phases, that saturation conditions of aluminium are easily attained, independently of the choice of $n_d(Al_2O_3)$.

Inspection of the the left panel of Figure 9 shows that the size reached by silicates (here represented by olivine, the dominant species) is unaffected by the choice of $n_d(Al_2O_3)$. The same holds for the fraction of silicon condensed into dust, which remains unchanged in the three cases discussed. This result indicates that the effects of the formation of alumina dust grains to the dynamics of the wind is negligible: the small amounts of $Al_2O_3$) dust formed (in turn due to the low abundances of aluminium with respect to the other species involved in dust formation) and the low extinction coefficients of this species render the effects of the radiation pressure negligible, leaving no opportunity to accelerate the wind, independently of the choice of the corresponding seed density. The conditions of the wind when entering the zone of solid carbon formation are practically independent of the formation of alumina dust in the more internal, which is the reason why the results regarding the size reached by

carbon grains and the fraction of carbon condensed into dust of the three cases discussed here are coincident.

## 7. Conclusions

We investigated the production of dust in the winds of AGB stars, by using a schematization for the growth of dust particles in the stationary wind of AGBs, applied to specific stages, selected along the evolutionary sequences of 1–8 $M_\odot$ stars.

The behaviour of dust production with the mass of the star is dichotomous. Low-mass stars with initial mass below $\sim$3–3.5 $M_\odot$ produce mainly carbonaceous dust, mostly composed by solid carbon, with minor contribution from silicon carbide; the stars with mass close to the threshold given above are characterized by the largest rates of dust production, as the surface of these stars becomes largely enriched in carbon during the AGB evolution. The higher mass counterparts produce silicates, with smaller quantities of alumina dust; the trend with mass is increasing also in this case, because the higher the mass of the star the higher the mass loss experienced, which favours dust production.

Dust production is found to be larger in higher metallicity environments. This is particularly evident in the case of massive AGBs, as the scarcity of silicon and aluminium renders dust production negligible in the metal-poor domain. In the case of carbon stars, dust forms in larger quantities in higher metallicity environments, because these stars assume a more expanded configuration than the lower metallicity counterparts, which again favours mass loss and dust formation. The SiC contribution to the overall dust formation in carbon stars is negligible in metal-poor environments.

The effects of drift on dust production are found to be important for drift velocities of the order of 5Km/s or more. The dust yields are generally larger in the models accounting for drift, whereas the effects on the optical depth are not significant, as the faster acceleration of the wind found in the models considering drift partly inhibits the formation of dust beyond the internal border of the dust formation zone. In the case of oxygen-rich stars the presence of drift favours the formation of pyroxene, which replaces olivine as the dominant dust species.

The release of the usual assumption that all the dust species grow on seed particles with the same number density leads to some differences with respect to the standard cases when carbon stars are considered. When the density of the seeds of SiC is assumed 10 times smaller than carbon, the SiC grains grow sufficiently big to affect the dynamics of the wind and accelerate the gas, with the consequence that the densities in the zone where solid carbon grains form are lower and the dust production rate is consequently reduced. In the case of carbon stars the choice of the number density

**Author Contributions:** Conceptualisation, S.T.; investigation, F.D., E.H.-M., P.V.; supervision, S.T.; writing—original draft, P.V. All authors have read and agreed to the published version of the manuscript.

**Funding:** P.V. benefited from the International Space Science Institute (ISSI, Bern, CH, and ISSI-BJ, Beijing, CN) thanks to the funding of the team "Chemical abundances in the ISM: the litmus test of stellar IMF variations in galaxies across cosmic time"; E.H.-M. acknowledges support from SECTEI (Education, Science, Technology and Innovation Counselor from Mexico city) postdoctoral fellowship.

**Conflicts of Interest:** The authors declare no conflict of interest.

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
