# Peer review of "Dust Formation in the Wind of AGB Stars—The Effects of Mass, Metallicity and Gas-Dust Drift"

_universe, doi:10.3390/universe8050270_

Round 1
Reviewer 1 Report
Review report on the manuscript:
Dust formation in the wind of AGB stars. The effects of mass, metallicity and gas-dust drift by S. Tosi et al.
This paper presents an interesting and extensive analysis of dust formation in the AGB phases of stellar evolution. It is well organized and very detailed. It presents useful results and it certainly deserves publication. From the scientific point of view I have only two (connected) key points that require clarification. They are important; and I invite the authors to dedicate some time to them.
1. Dust formed in AGB stars contributes to the inventory of presolar grains recovered in pristine meteorites and I would expect that this is commented upon. Model results should be at least broadly compared with the evidence from presolar grans. In particular, C-based dust (SiC) from masses 1.5-2.5 Mo is believed to account for the so-called "mainstream" SiC grains. This assumption looks in perfect agreement with what the authors actually find, but is not mentioned. I think the paper would gain enormously from a discussion of this point, which to me seems to be a direct confirmation of the results presented.
2. On the other hand, the authors find a relatively minor contribution to O-rich dust from very low masses; but these stars should be at the origin of presolar Al2O3 grains recovered in meteorites. These grains clearly show the effects of non-standard stellar mixing, in particular in their C and O isotopic ratios. I would not consider a remarkable weakness the lack of these processes in full stellar models (even now, they are usualy parameterized and there is ample motivation to discard them in the present manuscript). But this fact must not be simply ignored. The authors should discuss this point motivating their choices (or lack of) on extra-mixing. Even more, they should say whether they consider a problem or not the fact that O-rich dust from very low masses is not abundant, in view of the existence of these presolar grains.
Another point, not strictly related to the scientific content, but that actually affects it, is the presence of many (generally minor) formal issues in the text. I got the impression that the first author may be a PhD student or very young researcher, so this can be well due to inexperience. Then I took the liberty of indicating below the points that would need attention either in the English form, or in the clarity of exposition. It will require time to clean everything: but I am convinced that the qualty of the paper is worth the effort and that a clear presentation can attract much more attention on this nice paper.
Apart from the above maybe fussy notes, the paper is a good piece of work. I will be happy to accept it as soon as these points are taken into account.
Below see my detailed comments on the text.
Abstract
l.5 mass --> stellar mass
l.5 carbonaceous dust --> that may appear surprising, before your detailed computations. The formation of C-based dust indeed refers to the upper part of the mass range, and/or to the final stages, forming C-stars. Here you might anticipate something of the discussion (see your Figure 1 where a 1.5 Mo becomes a C-star only at the end; and your comments at page 8 and 10). Although C-based dust remains the most abundant component, it's better to clarify a little (see later my comment on section 3).
l.6 and 11 drift --> drifts? Otherwise an article?
l.10 "formation" repeated twice. Change the second into accumulation?
l.15 which inhibits--> and inhibits (otherwise the subject becomes "grains", which is plural)
Section 1. Introduction
l.1 The stars --> Stars
1st par.: A reference for the mid-IR shift of the spectral distribution is needed.
2nd and last par.: in the wind --> in the winds?
4th par.: popII --> population II (pop.II). Please define all the terms used.
last par.: drift --> drifts; otherwise put an article
Section 2.
2nd par.: allows considering --> allows us to consider
3rd par.: the value of the solar metallicity needs a reference (Asplund? Lodders?)
4th par.: -1 --> \simeq -1 (more precisely -0.85)
- popII stars --> I would say -0.85 is more typical of the oldest disc population, than of bona-fide popII stars. If you prefer your definition, this is ok with me, but then please specify the values you consider of pop. II against what you consider disc population
- were started from the pre-MS phase until.. --> please check the form. You cannot say something starts "until". It starts from... and proceeds until...
equation 1. Motivate the stationarity, e.g. with a reference. You are excluding
any modulation of the wind (i.e. the partial derivative versus time), which is ok, but must be commented at least once.
equation 2. Specify the units of k.
discussion of eq. 4: it's awkward to see lower-case letters after a full stop.
Say instead; "The parameters n_{d,i} and n_H are...respectivey, while epsilon_i...
Later, same par. "overcome the Fe components" (plural)
Final line of the section: "required ... ," --> "this last required"
etc. with similar polishing of the text in subsequent sections.
In general also:
- after a comma you need "which", not "that";
- "to allow" requires the object complement. I'll assume this
will be taken care of from now on, but pay attention.
- composite terms made of more than two words need hyphens
(e.g. "In carbon-rich environments", at page 8). You can skip
the hyphen if the first word is an adjectve (e.g. solid carbon
grains, again page 8).
Section 3.
In discussing hbb, please comment on the ensuing 12C/13C ratio, which is
a distinguished feature. Also, can you plot 13C in Figure 2.
Somewhere here or when speaking of low mass AGB stars, you should
mention that O-rich solids found in presolar materials are attributed to very low mass stars thanks to their isotopic ratios (mainly of O and C). It appears you did not consider deep mixing processes, or you did not discuss them.
Motivate this clearly (see my general comments at the beginning).
subsection 3.3, third par.
"as the in the lowest masses" --> skip the first "the". And check the form.
4th par. lower metallicities stars --> metallicity. Again, revise the form carefully!
Section 4, page 10.
Why is silicon scarce in the envelopes of low metallicity stars? To
me this is surprising. Silicon comes from core-collapse supernovae,
so that at lower-than-solar metallicities it should be enhanced with respect to Fe. I would expect that it is relatively more abundant,
not more scarse. Where and why am I wrong? Please explain.
Page 10, third paragraph. Specify the peculiarity of the very low masses, as
mentioned before, in connection to O-rich presolar grains
Page 11 last par. of the section.
"The dust produced by metal-poor... is smaller"? I guess is the dust "mass"
that is smaller
Section 5.
"the effects of drift velocity w of dust grains with respect to the gas". Put
an article ("a" drift velocity..)
page 12 after fig. 7 "drift" --> "a drift". Everywhere, check the use of singular
words, needing an article, with respect to plural terms.
page 13. What is the definition of the opical depth tau_10? I guess it is the one at 10 um, but it is not yet defined. Use a formula to define it.
page 13 4th par (and elsehwre, too). The term "condensation" it is not really correct. In chemistry, condensation is defined as the removal of heat from a system in such a way that a vapour is converted into a liquid. It's not something related to solid formation, despite the use by astronomers.
Are you referring to crystallization? Elsewhere you speak of monomers: are you meaning formation of monomer/polymer solids? Please specify.
Immediately later: "coexisting with and iron component". I guess you mean "an" iron component.
Section 7.
We inestigate --> We investigated
Author Response
We thank the competent referee for the careful reading of the manuscript.
We give below our reply to the main comments and to the minor points, with
requests of clarifications.
R) This paper presents an interesting and extensive analysis of dust formation in the AGB
phases of stellar evolution. It is well organized and very detailed. It presents useful
results and it certainly deserves publication. From the scientific point of view I have
only two (connected) key points that require clarification. They are important; and I
invite the authors to dedicate some time to them.
1. Dust formed in AGB stars contributes to the inventory of presolar grains recovered in
pristine meteorites and I would expect that this is commented upon. Model results should
be at least broadly compared with the evidence from presolar grans. In particular, C-based
dust (SiC) from masses 1.5-2.5 Mo is believed to account for the so-called "mainstream"
SiC grains. This assumption looks in perfect agreement with what the authors actually
find, but is not mentioned. I think the paper would gain enormously from a discussion of
this point, which to me seems to be a direct confirmation of the results presented.
A) We added a paragraph in section 4, where we discuss the point raised by the referee,
regarding the "mainstream" component.
R) 2. On the other hand, the authors find a relatively minor contribution to O-rich dust from
very low masses; but these stars should be at the origin of presolar Al2O3 grains
recovered in meteorites. These grains clearly show the effects of non-standard stellar
mixing, in particular in their C and O isotopic ratios. I would not consider a remarkable
weakness the lack of these processes in full stellar models (even now, they are usualy
parameterized and there is ample motivation to discard them in the present manuscript).
But this fact must not be simply ignored. The authors should discuss this point motivating
their choices (or lack of) on extra-mixing. Even more, they should say whether they
consider a problem or not the fact that O-rich dust from very low masses is not
abundant, in view of the existence of these presolar grains.
A) We apology with the referee for not being sufficiently clear on this side.
We did not mean that the production of oxygen-rich dust is null in low-mass AGBs,
rather that it is smaller in the massive AGBs, experiencing hot bottom burning.
We softened this sentence in section 4 and we added a further comment on the
possibility that low-mass, oxygen-rich AGBs are manufacturers of alumina dust.
We also commented on the choices regarding extra-mixing and the lack of the physical
mechanisms mentioned by the referee in section 2 (third paragraph).
R) Another point, not strictly related to the scientific content, but that actually affects
it, is the presence of many (generally minor) formal issues in the text. I got the
impression that the first author may be a PhD student or very young researcher, so this
can be well due to inexperience. Then I took the liberty of indicating below the points
that would need attention either in the English form, or in the clarity of exposition.
It will require time to clean everything: but I am convinced that the qualty of the paper
is worth the effort and that a clear presentation can attract much more attention on
this nice paper.
A) We are very grateful to the referee for the careful reading of the manuscript.
All the corrections suggested by the referee were considered and implemented into the
manuscript. We also added to the text all the references required by the referee.
In the following we give the reply only to the requests of clarifications asked by the referee.
All the other indications by the referee were considered, and the paper changed accordingly
R) l.5 carbonaceous dust --> that may appear surprising, before your detailed computations.
The formation of C-based dust indeed refers to the upper part of the mass range, and/or to
the final stages, forming C-stars. Here you might anticipate something of the discussion
(see your Figure 1 where a 1.5 Mo becomes a C-star only at the end; and your comments at
page 8 and 10). Although C-based dust remains the most abundant component, it's better
to clarify a little (see later my comment on section 3).
A) we changed the abstract to follow the referee's indications
R) equation 1. Motivate the stationarity, e.g. with a reference. You are excluding
any modulation of the wind (i.e. the partial derivative versus time), which is ok, but
must be commented at least once.
A) We added a motivation for the possibility of using the stationary approximation in section
2 (5th paragraph in the current version of the manuscript)
R) Section 4, page 10.
Why is silicon scarce in the envelopes of low metallicity stars? To me this is surprising.
Silicon comes from core-collapse supernovae,
so that at lower-than-solar metallicities it should be enhanced with
respect to Fe. I would expect that it is relatively more abundant,
not more scarse. Where and why am I wrong? Please explain.
A) We apology with the referee for this misunderstanding. In terms of the overall metal
content silicon is higher in the low-metallicity domain with respect to the solar
case. On the other hand, what matters for the formation of silicates is the
global silicon mass fraction, which is smaller the lower the metallicity of the star
Reviewer 2 Report
This is an interesting paper that seeks to connect stellar evolution models with simple dust formation models. It makes important predictions about the C and O AGB phases and the effect of drift velocity.
This is a complex problem, and the authors are forced to make various approximations. These should be more clearly stated including the approximations used by those whom they cite.
In particular they must comment on
- the lack of a size distribution of dust grains at a give radius. There is an assumption that the grains have a single size at a given radius.
- The dust formation model itself – do you use a chemical schema or something else
- How the approximations could affect the results
Also a review of observational evidence for C and O-type AGB stars, their masses and evolutionary tracks must be given. As it stands the results have no real context. This is important and should occupy a paragraph in the Introduction. Do the observational data support the dichotomy?
The predictive nature of these results would be enhanced by connecting this to a dust RT transfer model, which would allow predictions that could be directly compared with data e.g. generating simulated colour-colour diagrams for comparison with observations. This could be mentioned.
The Figure caption for Figure 7 seems to have little to do with Figure 7 itself.
The font sizes for the Figures are too small. I can only read them with a zoom factor of 150%
Author Response
We are grateful to the referee for the careful reading of the manuscript. Below our reply to the points raised in the report
R) These should be more clearly stated including the
approximations used by those whom they cite.
In particular they must comment on
* the lack of a size distribution of dust grains at a give radius. There is an assumption
that the grains have a single size at a given radius.
* The dust formation model itself – do you use a chemical schema or something else
* How the approximations could affect the results
A) To follow the referee's indications, we added in section 2 a discussion of the
approximations behind the dust formation modelling used in the present work, and how these could affect the results obtained.
R) Also a review of observational evidence for C and O-type AGB stars, their masses and
evolutionary tracks must be given. As it stands the results have no real context.
This is important and should occupy a paragraph in the Introduction. Do the observational
data support the dichotomy?
A) We added some references of recent works, where the results from stellar evolution
and dust formation modelling, that eventually produced synthetic photometry and spectral
energy distribution, are compared with the observations of stars in the Large Magellanic
Cloud.
R) The predictive nature of these results would be enhanced by connecting this to a dust
RT transfer model, which would allow predictions that could be directly compared with
data e.g. generating simulated colour-colour diagrams for comparison with observations.
This could be mentioned.
A) The referee is correct. This is connected with the point above and was added in the introduction
R) The Figure caption for Figure 7 seems to have little to do with Figure 7 itself.
A) We thank the referee for stressing this. We changed the caption of figure 7.
R) The font sizes for the Figures are too small. I can only read them with a zoom factor
of 150%
A) We agree with the referee on this. The font sizes of all the figures have been
increased.
Reviewer 3 Report
This manuscript is devoted to study the formation of dust in the wind of AGB stars, in which a stationary wind model is applied to the results from stellar evolution modeling to explore dust formation in the wind of stars evolving through the asymptotic giant branch. The role of mass and chemical composition in the dust production process is determined by considering stars of various masses. The work is written with suitable reasoning but there are some points that should be addressed properly by the author.
- In the literature, a lot of work has been done to discuss the dust formation of stars. In the introduction, the author should provide the references regarding his work in a better way. It should consist of better motivations in the extended form.
- In section 2. Author recommended different publications, which he opted in order to arrange this manuscript. He should discuss about the rationalities of taking models from these references. And the obtained results should be compared with them as well.
- In the graphical representation of the results particular values have been taken by the author. What is the reason for choosing these particular values? Why the considered values of time variation are very large in the absence of logarithmic scale.
Author Response
We thank the referee for the careful reading of the manuscript. Below our replies to the points raised in the report
R) This manuscript is devoted to study the formation of dust in the wind of AGB stars, in
which a stationary wind model is applied to the results from stellar evolution modeling
to explore dust formation in the wind of stars evolving through the asymptotic giant branch.
The role of mass and chemical composition in the dust production process is determined by
considering stars of various masses. The work is written with suitable reasoning but there
are some points that should be addressed properly by the author.
* In the literature, a lot of work has been done to discuss the dust formation of stars.
- In the introduction, the author should provide the references regarding his work in a
better way. It should consist of better motivations in the extended form.
A) We added in the introduction a list of references, to follow the referee's suggestion
R) In section 2. Author recommended different publications, which he opted in order to
arrange this manuscript. He should discuss about the rationalities of taking models from
these references. And the obtained results should be compared with them as well.
In the graphical representation of the results particular values have been taken by the
author. What is the reason for choosing these particular values? Why the considered values
of time variation are very large in the absence of logarithmic scale.
A) Regarding the models discussed in section 2, likely the referee refers to stellar
evolution modelling and the description of dust formation. As far as stellar evolution
modelling is concerned, we rely on our own models, obtained by means of the code
ATON, used in our research institute. For what concerns dust formation, the method
proposed by the Heidelberg group, discussed in section 2, is the only moment that at
the moment can be interfaced easily with results from stellar evolution modelling.